# A Scoping Review on Occupational Noise Mitigation Strategies and Recommendations for Sustainable Ship Operations

**DOI:** 10.3390/ijerph21070894

**Published:** 2024-07-09

**Authors:** Kresna Febriyanto, Joana Cristina Cardoso Guedes, Luis João Rodrigues Das Neves Correia Mourão

**Affiliations:** 1Faculty of Engineering, University of Porto, 4200-355 Porto, Portugaljccg@fe.up.pt (J.C.C.G.); 2Faculty of Public Health, Universitas Muhammadiyah Kalimantan Timur, 75124 Samarinda, Indonesia; 3CIETI/NBIN, Instituto Superior de Engenharia do Porto (ISEP), Polytechnic of Porto, 4249-015 Porto, Portugal; lnm@isep.ipp.pt

**Keywords:** workplace noise, control, prevention, vessel, boat, sailors, seafarers

## Abstract

Environmental and occupational noise has the potential to result in health risks. The presence of high noise levels aboard ships can cause substantial hazards that affect the well-being of those employed in the maritime industry. The study and implementation of occupational noise reduction aboard ships are of the highest priority for ensuring the well-being of marine workers, compliance with regulatory standards, protection of the environment, and improvement of overall operational efficiency and safety within the maritime sector. A scoping study was conducted to collect and summarize the existing scientific literature about approaches to preventing occupational noise in vessel operations. We searched electronic databases for papers published up to June 2024. Initially, 94 articles were identified for screening, and the present research produced 16 studies, which were finally analyzed. Resultantly, noise control may begin with elimination, substitution, engineering, administrative, and hearing protection (ear plugs or muffs). Noise control innovation would be started with engineering techniques. Hearing protection devices (HPDs) could be used to reduce noise and as an instrument of communication between sailors. More research needs to be carried out in order to find the best ways for maritime vessels to reduce noise at work and to see how well they work in lowering the risks that come with noise for workers on board.

## 1. Introduction

Globally, workplace noise is a widespread problem [1]. In the maritime sector, noise pollution has become a new emerging environmental issue caused by ships [2]. There are numerous types of noise sources on ships, including engine, ventilation, and compressor noise [3]. Environmental and occupational noise is a common nuisance that negatively impacts employees’ health [4].

Noise exposure is one of the most prevalent occupational risk factors associated with hearing loss. The Occupational Safety and Health Administration (OSHA) estimates that hearing loss disability costs $242 million annually [5]. Hearing impairment can be detected early before permanent hearing loss results from continued exposure [6]. This impairment makes it difficult to communicate with the surrounding environment and comprehend the speech of others, particularly in noisy environments. Consequently, it can also increase the risk of workplace accidents [7,8,9].

On the other hand, noise at work affects the health of sailors [1,10]. The navy, fishermen, passenger vessels (such as fast boats, ferry boats, etc.), oil and coal transport ships, and ships carrying tourists are at risk of noise exposure. A study reported that marine engineers remained the occupation with the highest incidence of hearing impairment [11]. Noise exposure can cause temporary threshold shifts (TTSs) and temporary hearing changes. It is believed that repeated TTSs without recovery is associated with noise-induced hearing loss (NIHL). Recovery from TTSs can take up to 24 h following exposure to loud noise, with longer exposure periods necessitating longer recovery periods [12].

Aside from auditory impairments, noise also has many health implications for sailors. According to Febriyanto et al. [10], the influence can be classified into two categories: physical problems (such as hearing loss and tinnitus, sleep disturbances, communication difficulties, poor concentration, dizziness, headaches, and fatigue) and psychological disorders (such as depression, anxiety, and stress).

An investigation conducted among navy personnel and fishermen revealed that they have sleep disruptions [13,14]. Another study conducted by Brooks and Greenberg [15] revealed that sailors who experience higher levels of noise and vibration are more likely to experience negative mental health effects. Stress, anxiety, and depression are categorized as psychological illnesses because they have a significant effect on an individual’s emotional and mental well-being [15,16]. The engine room aboard a vessel is going to show the highest levels of noise in the work environment, with the engine control room and other distant regions of the ship’s engine following suit [17].

Two additional noise exposure limits in vessels are provided by the National Institute for Occupational Safety and Health (NIOSH) and the International Maritime Organization (IMO) [12]. Following the IMO standard for protecting individuals from harmful noise pressure levels aboard ships, the permissible noise exposure is 85 dBA in work environments, 75 dBA in the engine control room, 65 dBA in the command and navigation space, and 60 dBA in restrooms. According to this standard, the cumulative exposure limit should not exceed 80 dBA per 24 h [6]. According to European Union Directive 2003/10/EC (2003) and The National Offshore Petroleum Safety and Environmental Management Authority (NOPSEMA) Regulations [18], “excessive noise” is defined as a level of noise that exceeds 85 dBA on an 8 h average of noise exposure. A threshold value for noise aboard ships has been determined, and several investigations have indicated that the intensity of noise often exceeds this limit [10,19].

While there are studies examining the effects of noise exposure on ships and its implications for human health, there is a lack of research dedicated to investigating occupational noise reduction strategies specifically developed for maritime vessels. In light of this, the purpose of this scoping review is to systematically find and summarize the current academic research strategies that are used to reduce occupational noise levels at work on ships. The outcomes of this study will provide valuable information for researchers, ship operators, and policymakers with an interest in improving occupational noise management within the maritime industry.

## 2. Materials and Methods

### 2.1. Study Protocol

The Preferred Reporting Items for Systematic reviews and Meta-Analysis extension for Scoping Reviews (PRISMA-ScR) methodology from Tricco, et al. [20] was used for this scoping review. The PRISMA-ScR methodology is employed as a systematic strategy for the identification and organization of relevant evidence based on primary themes, ideas, and existing knowledge.

### 2.2. Eligibility Criteria

The literature searches were limited to involving only full-text peer-reviewed papers accessible in English up to June 2024. Articles undergoing the publishing process, review papers, letters to the editor, opinions, conference proceedings, and book chapters were excluded. This study included papers that addressed noise control or prevention techniques aboard ships and focused on marine professionals, including those from the navy, fishermen, crew members, and other sailors.

### 2.3. Information Search and Research Strategies

The literature search included the following electronic databases: Scopus, Science Direct, Web of Science, Pubmed, and Ebsco Host. Different combinations of words were chosen using ‘AND’ and ‘OR’ operators to identify articles relevant to the study. In each database, keywords were combined as follows: (“occupational noise”) AND (ship OR boat OR vessel) AND (control OR prevention OR mitigation).

### 2.4. Study Selection Process

First, a title and abstract screening phase was followed by a full-text review technique as part of the review process. An independent investigation was used to carry out the initial screening of each article’s title and abstract. A full-text review was conducted for any article that the reviewer determined to be relevant. In the following phase, the investigator independently reviewed the entire text of the publications using the inclusion and exclusion criteria. Should a full-text examination yield any conflicting papers, the article will undergo an additional evaluation, and any remaining disagreements over its suitability will be deliberated with other investigators until a comprehensive agreement is reached.

### 2.5. Data Extraction and Synthesis of Results

The research group developed a method for collecting data from instruments. The purpose of this work was to validate the suitability of the study and provide a framework for extracting relevant study features. The data obtained consist of several specific details, including, but not limited to, the author’s identity and the year of publication, the type of ship under investigation, and the recommendation for noise control. Subsequently, the data obtained from the independent reviewer were deliberated over by the writers in order to ascertain the coherence and precision of the data. The process of data validation and coding was conducted by consolidating all extracted data into a Microsoft Excel spreadsheet. The objective of this scoping assessment is to present a comprehensive analysis of the various recommendations employed for mitigating occupational noise aboard ships.

## 3. Results

The literature search was conducted up to a publication date of June 2024 in the databases of Scopus, Science Direct, Web of Science, Pubmed, and Ebsco Host. The search returned 16, 5, 14, 47, and 12 documents containing the searched keyword (Figure 1). We eliminated duplicate results, opinion articles, conference proceedings, and review articles. Then, irrelevant articles were eliminated through abstract reading and full-text scanning. Also removed were studies unrelated to occupational noise control on ships. A total of 16 articles were reviewed to examine noise control strategies on ships.

According to the purpose of this investigation, the sixteen articles discuss the recommendation of ship noise control (Table 1). Most of the sixteen articles examined fishing vessel noise (seven papers) [12,21,22,23,24,25,26]. Other articles investigated ferryboats (three articles) [1,4,6], navy ships (three articles) [3,27,28], floating storage and offloading vessels (FSO) types (one paper) [18], and dredging ships (also one manuscript) [29]. The research on the ship recycling industry resulted in the publication of another article [30]. This research includes criteria despite the fact that they did not explicitly examine noise on board the vessel. The authors want to see noise control efforts from the shipbuilding process, during ship sailing, and when the ship is being repaired, resulting in very complex conclusions.

The ILO, IMO, and NIOSH [31] recommendations inform the noise control efforts on ships described by each author. Overall, noise control is achieved by using personal protective equipment (in the form of ear plugs or muffs).

## 4. Discussion

The most prevalent occupational hazard in the world is noise exposure [31]. Ships are the source of a new environmental problem involving noise pollution [2]. It is common knowledge that noise exposure can result in noise-induced hearing loss (NIHL) and is likely to cause non-auditory health effects [32]. Additionally, noise exposure has been associated with an increased accident risk [27], and physical and psychological disorders [10].

Multiple studies conducted on occupational noise aboard ships have demonstrated that the noise level is above the established limit [1,17,33,34]. Sound levels over 85 dB have detrimental effects on human health. The impact of exposure relies on elements such as the length of time, how often it occurs, and other important risk factors. Additionally, the impact may be influenced by characteristics such as gender, ethnicity, physical condition, and other causative agents originating from physical, chemical, biological, and other sources [34].

According to reports, one of the most hazardous occupations is fishing in many countries. Studies have revealed that fishermen suffer from numerous health issues, including hearing problems [22]. In addition, noise exposure has been an issue for naval personnel at sea for decades. Noise has been outlined as the most common occupational health hazard in the United States Marine corps, and a high rate of hearing impairment has been observed. However, only a few studies have reported sea noise levels in recent years, with most of these conducted on fishing vessels and container ships [27].

There are numerous international regulations governing noise on fishing vessels, including OSHA, NIOSH, IMO, and the United States Coast Guard (USCG). The USCG and OSHA noise exposure should not exceed 90 dBA for 8 h, 87 dBA for 12 h, and 82 dBA for 24 h. According to NIOSH and IMO, the noise exposure duration should not exceed 85 dBA, 83 dBA, and 80 dBA for 8 h, 12 h, and 24 h, respectively [12].

The International Maritime Organization’s (IMO) Code on Noise Levels on Board Ships, established in 2014, outlines the regulations for measuring noise aboard ships [35]. Typically, noise levels aboard a ship may be measured using sound level meters and dosimeters. To determine the level of noise on board, we identify ISO 9612 [35] for task-based measurements (TBMs), job-based measurements (JBMs), and full-day measures (FDMs) [19].

Occupational safety and health professionals utilize the Hierarchy of Control to determine how to implement practical and effective controls. This strategy classifies the framework for minimizing or removing noise hazards: elimination, substitution, engineering, administrative, and the wearing of personal protective equipment (PPE) [31]. OSHA and NIOSH standards require hearing protection devices (HPDs) for time-weighted average (TWA) exposures of 90 and 85 decibels, respectively. In contrast, USCG and IMO regulations mandate HPDs greater than 85 dBA regardless of exposure duration [12].

### 4.1. Noise Elimination

In the majority of instances, it is preferable to eliminate the source of dangerous noise [31]. However, this is impossible because the noise originates from the ship’s primary component, the propulsion engine. Without the machine, the boat cannot automatically move. In the result of the articles that have been reviewed, only one report states that elimination can be carried out in noise control on ships. The author does not mention that elimination is possible, but this can be achieved during the design phase by substituting them, purchasing low-noise equipment [18], or investing in the purchase of new ships [1].

### 4.2. Noise Subtitution

When elimination is not practicable, replacing loud equipment with quieter material might be the best method for protecting employees from dangerous noise [31]. Rutkowski and Korzeb [18] note that the use of quieter activities or processes without vibroacoustic impacts, such as surrogate surface preparation techniques, and low sound pressure level (SPL) engines [6], can reduce vibrational noise [24].

To mitigate the noise generated by fans, many approaches can be employed: substituting the current fans with aerodynamic, low-noise models; incorporating variable-speed drives into the fans; and adjusting the fan speed according to the specific ventilation needs at any given moment (a lower fan speed results in less noise) [18]. Changing the motors of the feed pump within the acoustic enclosure of the reduction gear lower casings and lubricating the pipe with oil on the dredging vessel can also contribute to noise reduction [29]. Zytoon’s research [22] on fishing vessels also indicated that substituting loud engines with comparatively quiet ones will undoubtedly decrease noise levels. The minimum cost for replacement might vary between $4000 and $7000, depending on the level of power needed.

Nevertheless, several factors must be taken into account, including whether replacing an old machine with a new one will increase costs, the cost of maintenance, and whether the new machine will perform better than the old one.

### 4.3. Engineering Control

The next stage is implementing engineering controls to avoid worker exposure to 8-h TWA excessive noise of 85 dB or above. Engineering controls involve physical work environment alterations, such as modifying equipment to remove noise sources and erecting noise-blocking barriers [31]. Controlling the noise hazard is typically accomplished by isolating workers from the hazard and erecting a barrier to block noise [12,18]. To reduce noise in small boats without decks and with outboard engines, two methods can be employed: insulating the engine cowling with materials that are not flammable to reduce sound, and performing periodic checks on the motors [22].

The application of engineering noise controls to shipboard environments is complicated by space and weight constraints, as well as flammability, sanitation, and accessibility requirements [12]. Nevertheless, several control measures could be implemented, such as the installation of an additional type of reflective muffler on the boat’s propulsion system [4], using sound-absorbing materials [1] with a high efficiency at the frequencies of 2000 and 4000 Hz on walls [6] in the engine room and silencers in the exhaust stack, machine enclosure, and isolation [6,22,23,30].

Silencers are frequently employed to minimize the engine intake and exhaust noise [36]. Depending on the specifications of the silencer, the engine’s horsepower, and the operational speed, silencers typically reduce noise by 10 to 40 dBA [37]. Additional engineering measures that can be implemented include the installation of acoustically absorbent materials (such as curtains, dividers, or panels), vibration dampeners, and treatments for walls and ceilings [12].

Furthermore, Burella and Moro’s [26] study discusses some successful methods for mitigating noise on ships. The engine room’s walls and ceiling should be insulated with mineral wool, and the doorway to the engine room should be adequately insulated, with any gaps covered. The installation of a floating floor is recommended directly above the engine room. Applying VEM-CLD to the surfaces that separate the engine room from the crew quarters is advised. VEM-CLD is an acronym that stands for viscoelastic materials in constrained layer damping configurations. Marine applications often employ this noise reduction technique to effectively decrease noise levels aboard vessels.

By relocating or isolating workers and other individuals from noise exclusion zones and places where noise levels exceed the exposure threshold [18], it is possible to incorporate design solutions to decrease hazardous noise levels on present and future vessels [1,24]. According to Rutkowski and Korzeb’s [18] research, this technique can be utilized to isolate employees and other individuals from noise exclusion zones and areas where noise levels exceed the exposure standard. All of these areas should be identified, and access to them should be restricted to individuals with adequate hearing protection. Exclusion zones should be marked with appropriate signs in accordance with safety signs for the workplace that alert workers and others of high noise levels and the requirement to wear hearing protection.

With the technological sophistication and advancements of the present day, it is possible to implement engineering control innovations, such as installing a sensor on the engine that can detect if it has been damaged or is not operating normally. With a detection sensor, it will be easier for workers to check through the monitor instead of checking directly. Thus, workers will be protected from noise exposure.

Numerous advancements in noise detection sensors have been made in recent years as a result of the research conducted in the past. An AVR-based microcontroller was used to create a noise detection instrument. The developed instrument has the ability to measure noise levels between 58 dBA and 95 dBA, and it can provide information in three conditions: secure, dangerous, and extremely dangerous, which is indicated by a warning lamp [38]. Then, Sayifah et al. [39] conducted microcontroller-based library noise detection. Based on their research, Rajagukguk and Sari [40] were able to draw the conclusion that the noise detection instrument based on an Arduino Uno and sensors with the tone decoder LM 567 performed admirably with a 0.7% error rate in measurement noise. With the instrument’s capabilities, the range of 50–100 dB values could be measured accompanied by alerts for three conditions: safety, noise, and peril.

### 4.4. Administrative Control

If engineering measures are impractical, companies may try guidelines, such as arranging to reduce exposure and ensure calm and convenient meal and break times [31]. Administrative controls typically include a crew training program [12,21,22,25]. According to Yadav et al.‘s research, fish harvesters said that they chose not to use hearing aids due to concerns about safety, such as difficulties in communication, the risk of going overboard, and the potential for accidents [25]. However, training is necessary to enhance workers’ understanding and awareness of the consequences by occupational noise [25] and the contributing elements that lead to it, as well as to encourage the adoption of hearing protection [30]. It is imperative that all stakeholders, including business organizations, the provincial government, and the federal government, offer supplementary training and education to improve the ship crew’s comprehension of the noise danger and its associated health consequences [25]. Srinoon et al. found that hearing protection device usage dramatically improved after the sailors completed the training program [28].

The installation of a warning sign [12,18,22,30], job rotation [12,18,22], reducing exposure by altering the time of activity, e.g., when noise-producing equipment is turned off, reducing the number of personnel exposed, and limiting the exposure duration [12,18,22] may also be carried out as an administrative change. Exclusion zones must be clearly marked with suitable signs that comply with safety regulations for the workplace. These signs serve to alert the crew and others about the presence of high noise levels and the requirement to wear hearing protection [18].

As a health measure, it is proposed that individuals undergo periodic medical examinations (every six months) [4,21,22,30] that include an audiometric test to determine the workers’ hearing impairment [6,21,22,23] at least once a year [23]. Even Esmaeili [6] indicates that personnel aboard these vessels are eligible for retirement or an early career change.

Among the innovations that can be considered for use in the work environment are the creation of occupational safety and health videos that can be played in the canteen/worker rest areas and the development of applications that can report workplace hazards. In this way, workers are able to report hazards they discover independently. In addition, it uses simulation techniques, such as video game consoles, to introduce new employees to the work environment and save time before the worker arrives at work. During orientation, the employees are provided with occupational health and safety information, including the hazards, risks, and precautions associated with the job [41].

### 4.5. Ear Protection

Over time, in a noisy environment, the issue of misunderstanding among employees has escalated into a pressing matter that requires prompt attention. Laborers are unaware of their surroundings due to loud noises, making co-ordination difficult. Finally, it will have an impact on the worker personally, such as health problems or work accidents. It is crucial to ensure the safety and well-being of workers by protecting them from excessive noise in the workplace, and it is equally necessary to prioritize effective communication throughout job activities. Consequently, there is a need for innovative ear coverings that may be used by ship personnel.

After all other alternatives for removing or decreasing noise at the source have been attempted, hearing protection devices such as earplugs and earmuffs should be made accessible to workers [3,22,23,24,25,27,30,31]. USCG recommends a hearing conservation program for fishing employees with exposures of 77 dBA or greater [12]. Additionally, NIOSH or OSHA recommend using HPDs while at work [27].

IMO assumes that earplugs and earmuffs can reduce noise by up to 20 and 30 decibels, respectively [22]. In the research conducted by Neitzel et al. [12], it was discovered that the average effective attenuation of hearing protection equipment employed in two commercial boats was 11 ± 5 dBA. An examination of ship recycling companies revealed an 11.5-decibel loss in sound intensity when earplugs were utilized, and a 12-decibel decrease in noise levels when earmuffs were worn. Moreover, the utilization of dual protection results in a reduction in noise up to a level of 17 dBA [30].

The ship’s crew used modern communication methods to overcome the challenge of limited communication capabilities. Effective communication is critical for preventing risks and alerting crews to possible hazards due to the disorganized and unplanned nature of ship activities [30]. They relied on a state-of-the-art crew communication system that allowed for unrestricted movement, hands-free operation, comprehensive communication, and protection for hearing. Speech communication headsets are utilized in a diverse array of applications. A communication headset typically comprises a set of headphones and a microphone affixed to the headset using a flexible boom that may be adjusted. A basic headset design features an open structure that allows for the minimal or no reduction in surrounding sounds. In headsets specifically built for loud surroundings, the headphones are placed into ear cups that are equipped with cushions to reduce noise [4]. HPDs are not only used to reduce noise exposure but also as a means of communication between workers [4,12,27,30].

Employees are more likely to utilize HPDs after receiving orientation training. The relationship between HPD usage and orientation training is significant due to the fact that HPD usage reflects safety behavior [41,42]. For many vessel operators, the stricter enforcement of the use of HPDs will likely be the most preferable means of reducing noise exposure. The dependence on HPDs to reduce exposure levels among workers must be accompanied by training on the correct fit and use of HPDs [43]. To enhance safety when working with hearing protection, it is crucial that we establish and enforce standardized protocols and safety inspections [30].

Overall, noise reduction on ships primarily targets the noise the engines produce. However, additional efforts are required on fishing boats to reduce noise, which originates not only from the engine but also from the fishing gear. The implementation of a dense rubber coating or mats on pots (used in crab, whelk, and lobster fisheries), sorting tables (used in crab, whelk, lobster, and shrimp fisheries), and deck floors helps reduce the noise caused by the gear. Additionally, sound-proofing enclosures are placed around electric or hydraulic generators on the deck or above the wheelhouse. Furthermore, improvements are made to the design of pots, net haulers, and net drums to incorporate a rubber coating on the drums [24].

Aside from the previously mentioned noise control measures involving elimination, substitution, engineering, administration, and ear protection, the ship owner can also undertake further efforts. This is achieved by implementing preventative maintenance on the engine through regular maintenance procedures. The measures implemented to mitigate noise pollution on trawling-type fishing vessels proved to be successful [22]. Despite this, implementing occupational noise control aboard a ship presents challenges due to the excessive cost, space limitations, and potential impact on onboard flammability [5].

## 5. Conclusions

This study concludes that noise control strategies can be implemented. The creative process begins with engineering control. The owner installs a sensor that can detect if the engine has been damaged or is not operating normally, as well as conduct automatic noise detection. Administrative controls are achieved by producing occupational safety and health videos, applications reporting workplace hazards, and simulations for new employees. HPDs are a tool for reducing noise and facilitating communication between workers. These objectives can be attained if the government adopts a regulatory approach that involves stakeholders to gain their support and encourage employees to behave safely on the job.

## Figures and Tables

**Figure 1 ijerph-21-00894-f001:**
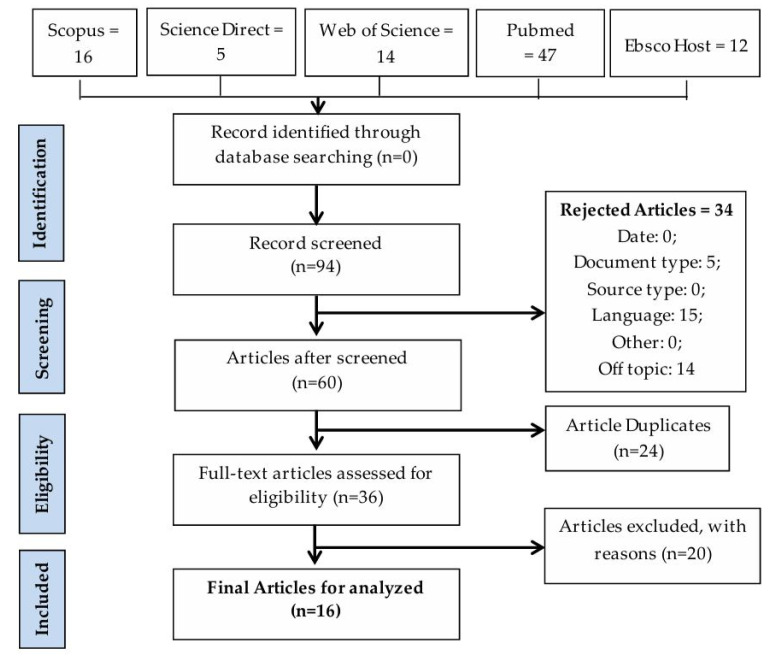
Flow diagram of article search results.

**Table 1 ijerph-21-00894-t001:** Grid synthesis.

No	Author, Year	Kind of Boat	Noise Recommendation Control
1	Stephen M. Bowes and Morton Corn, 1990 [29]	US dredging ship	Substitution: changing the motors of the feed pumps motors by acoustic enclosure of reduction gear lower casings and lubricating oil piping. Engineering: installing an acoustically isolated enclosed operating station (EOS) on the operational flat. Personal equipment: personal hearing protection.
2	Richard L. Neitzel et al., 2006 [12]	US fishing vessel	Engineering: isolation (use of acoustic barriers: curtains, partitions, or panels), vibration dampers, and wall and ceiling treatments. Administrative: reduced work hours, limited work periods in specific regions, or employee rotation and training. Personal equipment: earplugs and earmuffs.
3	Michele C. Paini et al., 2009 [21]	Brazilian fishing boat	Administrative: dissemination of information regarding the regulation and management of noise emissions produced by boat engines, and audiological care.
4	Aleksandar Nikolic and Emilija Nikolic, 2013 [4]	Britain ferryboat	Engineering: install an additional type of reflective muffler on the ferryboat’s propulsion system. Administrative: as a health measure, it is proposed that individuals undergo periodic medical examinations (every six months). Personal equipment: wear personal protective equipment (ear plugs or muffs), and utilize modern means of communication (wireless communication sets).
5	Mohamed A. Zytoon, 2013 [22]	Egyptian fishing vessels	Substitution: replacement of the engine to reduce noise. Engineering: control measures, such as using sound-absorbing materials in the engine room and silencers in the exhaust stack, and machine enclosure/isolation. Administrative: noise monitoring aboard fishing vessels, a safety training program, job rotation, and periodic medical examinations and audiometry. Personal equipment: using earplugs or ear muffs will protect your hearing.
6	Erlend Sunde et al., 2015 [27]	Royal Norwegian Navy vessel	Personal equipment: earplugs and earmuffs.
7	Gurmail S. Paddan, 2015 [3]	UK warship	Personal equipment: earplugs and earmuffs.
8	Rafet Emek Kurt et al., 2017 [30]	Ship recycling industry	Administrative: safety training, exclusion zones, standard procedures and safety checks, and monitoring health surveillance. Engineering: create low-cost, practical engineering controls. Personal equipment: effective and usable hearing protection.
9	Evelyn J. Albizu et al., 2020 [23]	Brazil fishing vessels	Administrative: audiometric test. Personal equipment: ear defenders.
10	Mihaela Picu and Laurentiu Picu, 2020 [1]	Romanian ferryboat	Engineering: encourage the use of sound-absorbing materials or invest in purchasing new ships.
11	Grzegorz Rutkowski and Jarosław Korzeb et al., 2021 [18]	UK floating storage and offloading (FSO) vessels	Elimination: this can be achieved during the design phase by substituting or purchasing low-noise equipment. Substitution: replace the existing fans with aerodynamic ones. Engineering: separate workers and other individuals from noise exclusion zones and areas; relocation or enclosure of loud equipment. Administrative: altering the timing of activities, decreasing the number of personnel at risk, and restricting the duration of exposure.
12	Giorgio Burella and Lorenzo Moro, 2021 [26]	Canadian fishing vessel	Engineering: insulating engine room walls, filling gaps in doorways, installing a floating floor above the engine room, and applying VEM-CLD to surfaces separating the engine room from crew quarters.
13	Giorgio Burella, Lorenzo Moro, and Barbara Neis, 2021 [24]	Canadian fishing vessel	Engineering: the development of effective design solutions to mitigate hazardous noise levels on existing and future vessels. Administration: programs to increase fish harvesters’ awareness of this significant occupational hazard and how to prevent injury. Personal equipment: the use of adequate hearing protection.
14	Reza Esmaeili et al., 2022 [6]	Iranian speed vessel	Substitution: used engines with low sound pressure level or controlling engine noise with highly effective sound insulation. Engineering: utilizing sound-absorbing materials with high efficiency on the walls of vessels. Administration: examining their hearing status, and retirement or an early job change.
15	Yadav et al., 2023 [25]	Canadian fish harvester boat	Engineering: vessel renovation or new vessel design. Administrative: provide training and education. Personal equipment: personal protective equipment.
16	Srinoon et al., 2023 [28]	Thailand navy ship	Administrative: hearing loss prevention programmes through websites to evaluate behavior in using hearing protection.

## Data Availability

The data availability statement is irrelevant to this study.

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
