# Peer review of "A Scoping Review on Occupational Noise Mitigation Strategies and Recommendations for Sustainable Ship Operations"

_ijerph, 2024, doi:10.3390/ijerph21070894_

Round 1

Reviewer 1 Report

Comments and Suggestions for Authors

Suggestions for additions to the article:

1.It would be useful to complete, update the statistics with publications also for 2023.

2.What are other databases for electronic search of scientific publications?

3.It would be possible to make a comparison with another database, e.g. Medline, PsycArticles, PsycInfo, SPORTDiscuss?

4.You note "increased risk of noise-related injuries"- please describe in more detail

5.Noise equipment on vessels what are they, please provide a diagram- I would recommend this area is described in more detail. 

6.Provide a concrete contribution-innovation for sustainable ship operation- with pictures or a more detailed description.

Author Response

Comments and Suggestions for Authors

Comments 1: It would be useful to complete, update the statistics with publications also for 2023.

Response 1: Thank you for your suggestion. We agree with this comment. Therefore, we have updated the data for review up to June 2024. There has been a lack of publishing papers available for review in the past two years (Only two papers published). Considering the limited research carried out on this topic. The updated review from literature can be found on page number 3 and 4 at section 2 (materials and methods) and section 3 (results).

Comments 2: What are other databases for electronic search of scientific publications?

Response 2: Thank you for pointing this out. Nevertheless, the study only relied on the following electronic databases, as stated in the research methods: Scopus, Science Direct, Web of Science, Pubmed, and Ebsco Host. No additional databases were utilised.

Comments 3: It would be possible to make a comparison with another database, e.g. Medline, PsycArticles, PsycInfo, SPORTDiscuss?

Response 3: Thank you for your inquiry this point. This study does not include any further information or comparisons with other databases. We conduct a thorough literature exploration exclusively using the specified database. The areas discussed in this research are to occupational safety and health, which is the field of study concerned with public health. The database employed is a comprehensive, multidisciplinary database that does not pertain exclusively to any particular discipline.

Comments 4: You note "increased risk of noise-related injuries"- please describe in more detail

Response 4: We appreciate your feedback. We have added a paragraph regarding this issue. The paragraph can be found at section 4, page number 6, second paragraph.

”Multiple studies conducted on occupational noise aboard ships have demonstrated that the noise level above the established limit [1, 17, 33, 34]. Sound levels over 85 dB have detrimental effects on human health. The impact of exposure relies on elements such as the length of time, how often it occurs, and other important risk factors. Additionally, the impact may be influenced by characteristics such as gender, ethnicity, physical condition, and other causative agents originating from physical, chemical, bio-logical, and other sources [34].”

Comments 5: Noise equipment on vessels what are they, please provide a diagram- I would recommend this area is described in more detail.

Response 5: We are thank for your point of view. We have included a paragraph to elucidate this particular topic. Nonetheless, this report argues for the recommendation to reduce occupational noise levels on the ship. According to the literature, the authors have already published an article discussing how to measure occupational noise on boats. More specifically, the methods, tools, and processes used to assess occupational noise levels on the ship will be detailed in further articles currently undergoing evaluation in other scientific publications. This added can be found at section 4 – page number 7.

“The International Maritime Organisation's (IMO) Code on Noise Levels on Board Ships, established in 2014, outlines the regulations for measuring noise aboard ships [33]. Typically, noise levels aboard a ship may be measured using sound level metres and dosimeters. To determine the level of noise on board, identify ISO 9612:2009 for task-based measurements (TBM), job-based measurements (JBM), and full-day measures (FDM) [19]”

Comments 6: Provide a concrete contribution-innovation for sustainable ship operation- with pictures or a more detailed description.

Response 6: We are thankful for your opinion. Out of the 16 papers that were examined, only three specifically addressed noise control on board and provided explanations of the most successful approaches (References No. 30, 26, and 28). Therefore, future studies must make a concerted effort to regulate noise on board and determine the most effective and efficient approach.

The explanation of reference number 26 can be found at section 4.3 on page 8

“Furthermore, Burella and Moro's [26] study discusses some successful methods for mitigating noise on ships. The engine room's walls and ceiling should be insulated with mineral wool, and the doorway to the engine room should be adequately insulated, with any gaps covered. The installation of a floating floor is recommended directly above the engine room. Applying VEM-CLD to the surfaces that separate the engine room from the crew quarters is advised. VEM-CLD is an acronym that stands for Viscoelastic Materials in Constrained Layer Damping Configurations. Marine applications often employ this noise reduction technique to effectively decrease noise levels aboard vessels.”

The detailed of reference number 28 can be found at section 4.4 on page 9

“Srinoon et al. found that hearing protection device usage dramatically improved after the sailors completed the training program [28].”

A Paragraph for reference number 30 you might found at section 4.5 on page 10

“IMO assumes that earplugs and earmuffs can reduce noise by up to 20 and 30 decibels, respectively [22]. In research conducted by Neitzel et al. [12], it was discovered that the average effective attenuation of hearing protection equipment employed in two commercial boats was 11±5 dBA. An examination of ship recycling companies revealed an 11.5-decibel loss in sound intensity when earplugs were utilized, and a 12-decibel decrease in noise levels when earmuffs were worn. Moreover, the utilization of dual protection results in a reduction of noise up to a level of 17 dBA [30].”

Reviewer 2 Report

Comments and Suggestions for Authors

In the manuscript entitled Implementation of A Scoping Review on Occupational Noise Mitigation Strategies and Recommendations for Sustainable Ship Operations, the authors call for more research to be done to find the best ways to reduce noise from work on vessels at sea and to see how well they are doing in reducing the risks posed by noise from workers on board. However, this manuscript also has some shortcomings and needs to be revised.

Comments 1: The description of the harm of noise in the introduction is verbose and repetitive. Most of the content is a repetitive and worthless description of the physical and mental health of the crewPlease adjust.

Comments 2: In the discussion, the specific measures described in the existing literature are simply listed, and there is no concise analysis of the correlation between the literature. 

Comments 3: Specific measures based on literature research do not well reflect the effectiveness of application in practice. Please improve it.

Author Response

Comments and Suggestions for Authors

Comments 1: The description of the harm of noise in the introduction is verbose and repetitive. Most of the content is a repetitive and worthless description of the physical and mental health of the crew. Please adjust.

Response 1: Thank you for pointing this out. We agree with this comment. We have expanded the introduction to include additional information on physical and mental health problem of the crew. This change can be found – page number 1 to 3 at introduction part.

Comments 2: In the discussion, the specific measures described in the existing literature are simply listed, and there is no concise analysis of the correlation between the literature.

Response 2: We are thank for your point of view. We adopted a risk management hierarchy for noise mitigation, which is based on the guidelines provided by OSHA and NIOSH. Occupational risk management has five stages: elimination, substitution, engineering, administration, and self-protection. As a result, we classified all of the publications we reviewed based on risk management. The current version of the paper provides more detailed description of each level of noise control, which may be found in sections 4.1 to 4.5.

Comments 3: Specific measures based on literature research do not well reflect the effectiveness of application in practice. Please improve it.

Response 3: Thank you for your inquiry this point. Among the 16 studies analyzed, only three focused on noise reduction on board and offered detailed descriptions of the effective methods (References No. 30, 26, and 28). The three references that describe effective approaches for the mitigation of noise on board have been outlined in the manuscript, which can be found in sections 4.3, 4.4, and 4.5.

Reviewer 3 Report

Comments and Suggestions for Authors

This Scoping Review did not take the earphone user into consideration for occupational or non-occupational noise exposure. Moreover, how to propose the noise mitigation strategies and recommendations for sustainable ship operations is an important issue in the future. It is suggested that the authors should search some articles about the hearing impact of the mobile phone or earphone users. Please consider the following suggestion.

1) To correct for "noise-induced hearing loss (NIHL)" in L48 and L180.

2) To correct the cited references consistant in this paper. (L216, L241, L254, L258)

3) The reference no.22 (L.376-378) was not found, please to confirm it.

Author Response

Comments and Suggestions for Authors

Comments 1: To correct for "noise-induced hearing loss (NIHL)" in L48 and L180.

Response 1: Thank you for pointing this out. We have changed intonoise-induced hearing loss (NIHL)”, and this change can be found on page number 2 paragraph 1, and page number 6 at the first paragraph of discussion section.

Comments 2: To correct the cited references consistant in this paper. (L216, L241, L254, L258).

Response 2: We are grateful for your suggestion. We have clarified the references in the revised manuscript. Please see the attached file.

Comments 3: The reference no.22 (L.376-378) was not found, please to confirm it.

Response 3: Thank you for your inquiry this point. The cited reference on the text can be found on page number 4 (section 3, before the table), page number 6 (the first discussion part), and page number 7-9.